# ConceptHash: Interpretable Hashing for Fine-grained Retrieval and Generation

## Abstract

Existing fine-grained hashing methods typically lack code interpretability as they compute hash code bits holistically using both global and local features. To address this limitation, we propose ConceptHash, a novel method that achieves sub-code level interpretability. In ConceptHash, each sub-code corresponds to a human-understandable concept, such as an object part, and these concepts are automatically discovered without human annotations. Specifically, we leverage a Vision Transformer architecture and introduce concept tokens as visual prompts, along with image patch tokens as model inputs. Each concept is then mapped to a specific sub-code at the model output, providing natural sub-code interpretability. To capture subtle visual differences among highly similar sub-categories (e.g., bird species), we incorporate language guidance to ensure that the learned hash codes are distinguishable within fine-grained object classes while maintaining semantic alignment. This approach allows us to develop hash codes that exhibit similarity within families of species while remaining distinct from species in other families. Extensive experiments on four fine-grained image retrieval benchmarks demonstrate that ConceptHash outperforms previous methods by a significant margin, offering unique sub-code interpretability as an additional benefit.

## 1 Introduction

Learning to hash is an effective approach for constructing large-scale image retrieval systems (Luo et al., 2020). Previous methods primarily use pointwise learning algorithms with efficient hash center-based loss functions (Su et al., 2018; Yuan et al., 2020; Fan et al., 2020; Hoe et al., 2021). However, these methods mainly focus on global image-level information and are best suited for distinguishing broad categories with distinct appearance differences, like apples and buildings. In many real-world applications, it's essential to distinguish highly similar sub-categories with subtle local differences, such as different bird species. In such scenarios, the computation of hash codes that capture these local, class-discriminative visual features, like bird beak color, becomes crucial.

Recent fine-grained hashing methods (Cui et al., 2020; Wei et al., 2021a; Shen et al., 2022) extract local features and then combine them with global features to compute hash codes. However, this approach lacks interpretability because hash codes are derived from a mix of local and global features. As a result, it becomes challenging to establish the connection between human-understandable concepts (e.g., tail length and beak color of a bird) and individual or blocks of hash code bits (sub-codes). These concepts are typically local, as globally fine-grained classes often share similar overall characteristics (e.g., similar body shapes in all birds).

The importance of model interpretability is growing in practical applications. Interpretable AI models boost user confidence, assist in problem-solving, offer insights, and simplify model debugging (Molnar, 2020; Lundberg & Lee, 2017; Van der Velden et al., 2022). In the context of learning-to-hash, interpretability pertains to the clear connection between semantic concepts and hash codes. For instance, a block of hash code bits or sub-code should convey a specific meaning that can be traced back to a local image region for visual inspection and human comprehension. While the methods introduced in previous works (Wei et al., 2021a; Shen et al., 2022) were originally conceived with interpretability in mind, they have made limited progress in this regard. This limitation stems from the fact that their hash codes are computed from aggregated local and global feature representations,

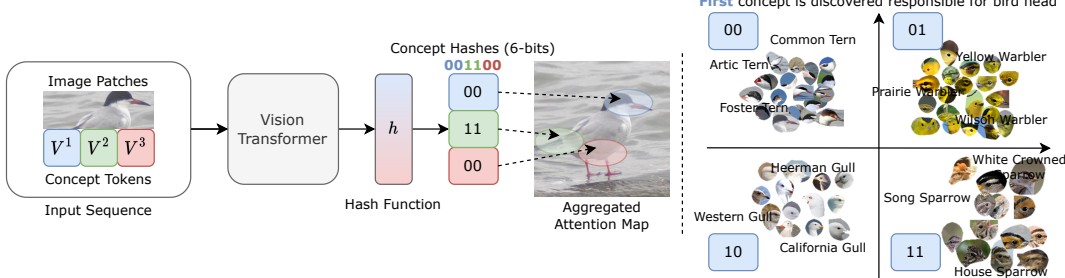

Figure 1: In the proposed ConceptHash, a set of concept tokens (3 tokens in this illustration) are introduced in a vision Transformer to discover automatically human understandable semantics (*e.g.*, bird head by the first concept token for generating the first two-bit sub-code **00**).

making it challenging to establish a direct association between a sub-code and a local semantic concept.

To address the mentioned limitation, we present an innovative concept-based hashing approach named *ConceptHash*, designed for interpretability (see Fig. 1). Our architecture builds upon the Vision Transformer (ViT) (Dosovitskiy et al., 2021). To enable semantic concept learning, we introduce learnable concept tokens, which are combined with image patch tokens as input to ViT. At the ViT's output, each query token corresponds to a sub-code. Concatenating these sub-codes yields the final hash code. Notably, the visual meaning of each concept token is evident upon inspection. This intrinsic feature makes our model interpretable at the sub-code level since each sub-code directly corresponds to a concept token. Additionally, we harness the rich textual information from a pretrained vision-language model (CLIP (Radford et al., 2021)) to offer language-based guidance. This ensures that our learned hash codes are not only discriminative within fine-grained object classes but also semantically coherent. By incorporating language guidance, our model learns hash codes that exhibit similarity within species' families while maintaining distinctiveness from species in other families. This approach enhances the expressiveness of the hash codes, capturing nuanced visual details and meaningful semantic distinctions, thereby boosting performance in fine-grained retrieval tasks.

Our **contributions** are as follows. **(1)** We introduce a novel ConceptHash approach for interpretable fine-grained hashing, where each sub-code is associated with a specific visual concept automatically. **(2)** We enhance the semantics of our approach by incorporating a pretrained vision-language model, ensuring that our hash codes semantically distinguish fine-grained classes. **(3)** Extensive experiments across four fine-grained image retrieval benchmarks showcase the superiority of ConceptHash over state-of-the-art methods, achieving significant improvements of 6.82%, 6.85%, 9.67%, and 3.72% on CUB-200-2011, NABirds, Aircraft, and CARS196, respectively

## 2 RELATED WORK

**Learning to hash.** Deep learning-based hashing (Xia et al., 2014; Lai et al., 2015; Wang et al., 2016b; Cao et al., 2017; 2018) has dominated over conventional counterparts (Indyk & Motwani, 1998; Gionis et al., 1999; Kulis & Grauman, 2009; Weiss et al., 2009; Kulis & Darrell, 2009; Gong et al., 2012; Kong & Li, 2012; Norouzi & Fleet, 2011; Norouzi et al., 2012). Recent works focus on a variety of aspects (Luo et al., 2020), *e.g.*, solving vanishing gradient problems caused by the sign function sign (Su et al., 2018; Li & van Gemert, 2021), reducing the training complexity from $O(N^2)$ to $O(N)$ with pointwise loss (Su et al., 2018; Yuan et al., 2020; Fan et al., 2020; Hoe et al., 2021) and absorbing the quantization error objective (Fan et al., 2020; Hoe et al., 2021) into a single objective. These works usually consider the applications for differentiating coarse classes with clear pattern differences (*e.g.*, houses vs. cars), without taking into account hash code interpretability.

**Fine-grained recognition.** In many real-world applications, however, fine-grained recognition for similar sub-categories is needed, such as separating different bird species (Wei et al., 2021b). As the class discriminative parts are typically localized, finding such local regions becomes necessary.

Typical approaches include attention mechanisms (Zheng et al., 2017; 2019b;a; Jin et al., 2020; Chang et al., 2021; Peng et al., 2017; Wang et al., 2018; Yang et al., 2022a), specialized architectures/modules (Zheng et al., 2018; Weinzaepfel et al., 2021; Wang et al., 2021; He et al., 2022; Behera et al., 2021; Lin et al., 2015; Zeng et al., 2021; Huang & Li, 2020; Sun et al., 2020), regularization losses (Du et al., 2020; Dubey et al., 2018; Chen et al., 2019; Sun et al., 2018; Chang et al., 2020), and finer-grained data augmentation (Du et al., 2020; Lang et al., 2022). They have been recently extended to *fine-grained hashing*, such as attention learning in feature extraction (Wang et al., 2020; Lu et al., 2021; Jin et al., 2020; Lang et al., 2022; Chen et al., 2022c; Xiang et al., 2021) and feature fusion (Shen et al., 2022; Wei et al., 2021a; Cui et al., 2020). However, in this study we reveal that these specialized methods are even less performing than recent coarse-grained hashing methods, in addition to lacking of code interpretability. Both limitations can be addressed with the proposed ConceptHash method in a simpler architecture design.

**Model interpretability.** Seeking model interpretability has been an increasingly important research topic. For interpretable classification, an intuitive approach is to find out the weighted combinations of concepts (Kim et al., 2018; Koh et al., 2020; Zhou et al., 2018; Stammer et al., 2022; Sawada & Nakamura, 2022; Wu et al., 2023; Yuksekgonul et al., 2022; Yang et al., 2022b) (a.k.a. prototypes (Rymarczyk et al., 2021; Nauta et al., 2021; Arik & Pfister, 2020)). This is inspired by human's way of learning new concepts via subconsciously discovering more detailed concepts and using them in varying ways for world understanding (Lake et al., 2015). The concepts can be learned either through fine-grained supervision (*e.g.*, defining and labeling a handcrafted set of concepts) (Zhang et al., 2018; Rigotti et al., 2022; Koh et al., 2020; Yang et al., 2022b), or weak supervision (*e.g.*, using weak labels such as image-level annotations) (Wang et al., 2023; Oikarinen et al., 2023), or self-supervision (*e.g.*, no any manual labels) (Alvarez Melis & Jaakkola, 2018; Wang et al., 2023).

In this study, we delve into the realm of semantic concept learning within the context of learning-to-hash, with a distinct emphasis on achieving sub-code level interpretability. While $A^2$-Net Wei et al. (2021a) has asserted that each bit encodes certain data-derived attributes, the actual computation of each bit involves a projection of both local and global features, making it challenging to comprehend the specific basis for the resulting bit values. In contrast, our approach, ConceptHash, takes a different approach. It begins by identifying common concepts (e.g., head, body) and subsequently learns the corresponding sub-codes within each concept space. Besides, our empirical findings demonstrate that ConceptHash outperforms previous methods in terms of performance.

**Language guidance.** Vision-language pretraining at scale (Radford et al., 2021) has led to a surge of exploiting semantic language information in various problems (Wang et al., 2016a; Yang et al., 2016; Chen et al., 2020; Kim et al., 2021; Li et al., 2021b; 2019; 2020). For example, text embedding has been used to improve dense prediction (Rao et al., 2022), interpretability (Yang et al., 2022b), metric learning (Roth et al., 2022; Kobs et al., 2023), self-supervised learning (Banani et al., 2023), and visual representations (Sariyildiz et al., 2020; Huang et al., 2021). For the first time, we explore the potential of language guidance for fine-grained hashing, under the intuition that semantic information could complement the subtle visual differences of sub-categories while simultaneously preserving similarity within species belonging to the same family.

## 3 METHODOLOGY

We denote a training dataset with $N$ samples as $\mathcal{D} = \{(x_i, y_i)\}_{i=1}^N$, where $x_i$ is the $n$-th image with the label $y_i \in \{1, ..., C\}$. Our objective is to learn a hash function $\mathcal{H}(x) = h(f(x))$ that can convert an image $x_i$ into a $K$-bits interpretable hash code $b \in \{-1, 1\}^K$ in a discriminative manner, where $f$ is an image encoder (*e.g.*, a vision transformer) and $h$ is a hashing function. To that end, we introduce a novel interpretable hashing approach, termed ***ConceptHash***, as illustrated in Fig. 2.

### 3.1 LEARNING TO HASH WITH AUTOMATICALLY DISCOVERED CONCEPTS

Given an image, our ConceptHash aims to generate an interpretable hash code composed by concatenating $M$ sub-codes $\{b^1, ..., b^M\}$. Each sub-code $b^m \in \{-1, 1\}^{K/M}$ expresses a particular visual concept discovered automatically, with $K$ the desired hash code length. To achieve this, we employ a Vision transformer (ViT) architecture denoted as $f$. At the input, apart from image patch tokens, we

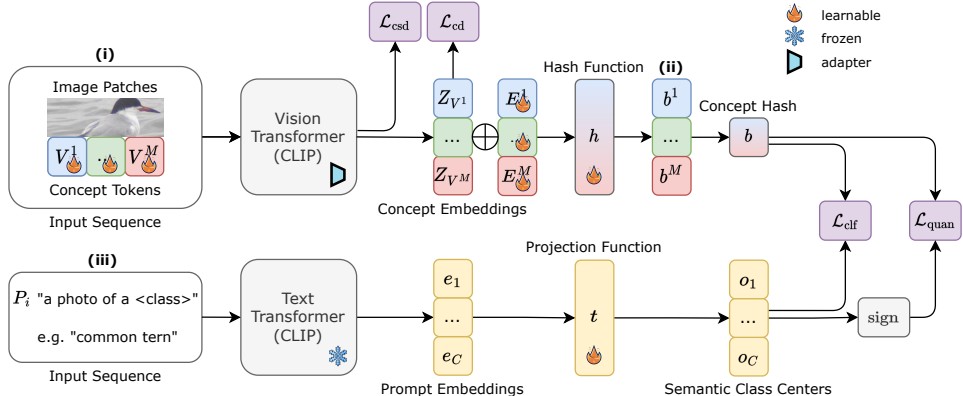

Figure 2: Overview of our ConceptHash model in a Vision Transformer (ViT) framework. To enable sub-code level interpretability, (i) we introduce a set of $M$ concept tokens along with the image patch tokens as the input. After self-attention based representation learning, (ii) each of these concept tokens is then used to compute a sub-code, all of which are then concatenated to form the entire hash code. (iii) To compensate for limited information of visual observation, textual information of class names is further leveraged by learning more semantically meaningful hash class centers. For model training, a combination of classification loss $\mathcal{L}_{\text{clf}}$, quantization error $\mathcal{L}_{\text{quan}}$, concept spatial diversity constraint $\mathcal{L}_{\text{csd}}$, and concept discrimination constraint $\mathcal{L}_{\text{cd}}$ is applied concurrently. To increase training efficiency, Adapter (Houlsby et al., 2019) is added to the ViT instead of fine-tuning all parameters.

introduce a set of $M$ learnable `concept tokens`:

$$Z^{(0)} = \text{concat}(x^1, ..., x^{\text{HW}}, [V^1], ..., [V^M]), \tag{1}$$

where concat denotes the concatenation operation, $[V^m]$ is the $m$-th concept token, $x^i$ is the $i$-th image patch token with $HW$ the number of patches per image (commonly, $HW = 7 * 7 = 49$). With this augmented token sequence $Z^{(0)}$, we subsequently leave the ViT model to extract the underlying concepts via the standard self-attention-based representation learning:

$$Z^{(L)} = f(Z^{(0)}) \in \mathbb{R}^{(HW+M) \times D},$$
$$\text{where } Z^{(l)} = \text{MSA}^{(l)}(Z^{(l-1)}), \tag{2}$$

in which $Z^{(l)}$ is the output of the $l$-th layer in a ViT and $\text{MSA}^{(l)}$ is the self-attention of $l$-th layer in $f$ (the MLP, Layer Normalization (Ba et al., 2016), and the residual adding were omitted for simplicity). The last $M$ feature vectors of $Z^{(L)}$ (denoted as $Z$ for simplicity), $Z[\text{HW+1:HW+}M]$, is the representation of the concepts discovered in a data-driven fashion, denoted as $Z_{[V^1]}, ..., Z_{[V^M]}$.

**Interpretable hashing.** Given each concept representation $Z_{[V^m]}$, we compute a specific sub-code $b^m$. Formally, we design a concept-generic hashing function as

$$b^m = \text{h}(Z_{[V^m]} + E_m), \quad b = \text{concat}(b^1, ..., b^M), \tag{3}$$

where $E_m \in \mathbb{R}^{M \times D}$ is the $m$-th concept specificity embedding that enables a single hashing function to be shared across different concepts. In other words, the concept specificity embedding serves the purpose of shifting the embedding space of each specific concept to a common space, allowing a single hashing function to be applied to all concepts and convert them into hash codes. Note that $b$ (the concatenation of all sub-codes) is a continuous code. To obtain the final hash code, we apply a sign function $\hat{b} = \text{sign}(b)$.

## 3.2 LANGUAGE GUIDANCE

Most existing fine-grained hashing methods rely on the information of visual features alone (Shen et al., 2022; Wei et al., 2021a; Cui et al., 2020). Due to the subtle visual difference between sub-categories, learning discriminative hashing codes becomes extremely challenging. We thus propose using the readily available semantic information represented as an embedding of the class names as an auxiliary knowledge source (*e.g.*, the semantic relation between different classes).

More specifically, in contrast to using random hash class centers as in previous methods (Yuan et al., 2020; Fan et al., 2020; Hoe et al., 2021), we learn to make them semantically meaningful under language guidance. To that end, we utilize the text embedding function $g(\cdot)$ of a pre-trained CLIP (Radford et al., 2021) to map a class-specific text prompt ($P \in \{P_c\}_{c=1}^C$ where $P_c = $ "a photo of a [CLASS]") to a pre-trained embedding space, followed by a learnable projection function $t(\cdot)$ to generate the semantic class centers:

$$e_c = g(P_c), \quad o_c = t(e_c). \tag{4}$$

The class centers $o = \{o_c\}_{c=1}^C \in \mathbb{R}^{C \times K}$ then serve as the hash targets for the classification loss in Eq. 6 and 7. This ensures that the learned hash codes are not only discriminative within fine-grained object classes but also semantically aligned. More specifically, the integration of language guidance guides the model to output hash codes that exhibit similarity within families of species while preserving discriminativeness from species belonging to other families (see Sec. 4.3 and Fig. 5).

### 3.3 LEARNING OBJECTIVE

The objective loss function to train our ConceptHash model is formulated as:

$$\mathcal{L} = \mathcal{L}_{\text{clf}} + \mathcal{L}_{\text{quan}} + \mathcal{L}_{\text{csd}} + \mathcal{L}_{\text{cd}}. \tag{5}$$

with each loss term as discussed below.

The first term $\mathcal{L}_{\text{clf}}$ is the classification loss for discriminative learning:

$$\mathcal{L}_{\text{clf}} = -\frac{1}{N} \sum_{i=1}^N \log \frac{\exp(\cos(o_{y_i}, b_i)/\tau)}{\sum_{c=1}^C \exp(\cos(o_c, b_i)/\tau)}, \tag{6}$$

where $\tau$ is the temperature ($\tau = 0.125$ by default), $C$ is the number of classes, and $\cos$ computes the cosine similarity between two vectors. This is to ensure the hash codes are discriminative.

The second term $\mathcal{L}_{\text{quan}}$ is the quantization error:

$$\mathcal{L}_{\text{quan}} = -\frac{1}{N} \sum_{i=1}^N \log \frac{\exp(\cos(\hat{o_{y_i}}, b_n)/\tau)}{\sum_{c=1}^C \exp(\cos(\hat{o_c}, b_i)/\tau)},$$
$$\text{where } \{\hat{o}_c\}_{c=1}^C = \{\text{sign}(o_c)\}_{c=1}^C. \tag{7}$$

Instead of directly minimizing the quantization error, we use the set of binarized class centers $\hat{o}$ as the classification proxy, which is shown to make optimization more stable (Hoe et al., 2021).

The third term $\mathcal{L}_{\text{csd}}$ is a concept spatial diversity constraint:

$$\mathcal{L}_{\text{csd}} = \frac{1}{NM(M-1)} \sum_{i \neq j} \cos(A_i, A_j), \tag{8}$$

where $A_i \in \mathbb{R}^{N \times HW}$ is the attention map of the $i$-th concept token in the last layer of the self-attention $\text{MSA}^{(L)}$ of $f$, obtained by averaging over the multi-head axis, The idea is to enhance attention map diversity (Weinzaepfel et al., 2021; Li et al., 2021a; Chen et al., 2022b), thereby discouraging concepts from focusing on the same image region.

The forth term $\mathcal{L}_{\text{cd}}$ is the concept discrimination constraint:

$$\mathcal{L}_{\text{cd}} = -\frac{1}{NM} \sum_{i=1}^N \sum_{m=1}^M \log \frac{\exp(\cos(\hat{W}_{y_i}, \hat{Z}_{[V^m]_i})/\tau)}{\sum_{c=1}^C \exp(\cos(\hat{W}_c, \hat{Z}_{[V^m]_i})/\tau)},$$
$$\text{where } \hat{Z}_{[V^m]_i} = Z_{[V^m]_i} + E^m, \tag{9}$$

where $\{\hat{W}_c\}_{c=1}^C \in \mathbb{R}^{C \times D}$ are learnable weights and $E \in \mathbb{R}^{M \times D}$ is the concept specificity embedding (same as $E$ in Eq. 3). The feature-to-code process incurs substantial information loss (i.e., the projection from $Z_{[V]}$ to $b$), complicating the optimization. This loss serves a dual purpose: promoting discriminative concept extraction and supplying additional optimization gradients.

Table 1: Comparing with prior art hashing methods. Note, ITQ is an unsupervised hashing method considered as the baseline performance. $^{\dagger}$: Originally reported results. **Bold**: The best performance.

| Dataset | | CUB-200-2011 | | | NABirds | | | FGVC-Aircraft | | | Stanford Cars | | |
|---|---|---|---|---|---|---|---|---|---|---|---|---|---|
| Method | | 16 | 32 | 64 | 16 | 32 | 64 | 16 | 32 | 64 | 16 | 32 | 64 |
| ITQ | Gong et al. (2012) | 7.82 | 11.53 | 15.42 | 3.40 | 5.50 | 7.60 | 8.12 | 9.78 | 10.87 | 7.80 | 11.41 | 15.16 |
| HashNet | Cao et al. (2017) | 14.45 | 23.64 | 32.76 | 6.35 | 8.93 | 10.21 | 20.36 | 27.13 | 32.68 | 18.23 | 25.54 | 32.43 |
| DTSH | Wang et al. (2016b) | 25.16 | 27.18 | 27.89 | 3.35 | 6.00 | 7.87 | 21.32 | 25.65 | 36.05 | 20.48 | 27.40 | 28.34 |
| GreedyHash | Su et al. (2018) | 73.87 | 81.37 | 84.43 | 54.63 | 74.63 | 79.61 | 49.43 | 75.21 | 80.81 | 75.85 | 90.10 | 91.98 |
| CSQ | Yuan et al. (2020) | 69.61 | 75.98 | 78.19 | 62.33 | 71.24 | 73.61 | 65.94 | 72.81 | 74.05 | 82.16 | 87.89 | 87.71 |
| DPN | Fan et al. (2020) | 76.63 | 80.98 | 81.96 | 68.82 | 74.52 | 76.75 | 70.86 | 74.04 | 74.31 | 87.67 | 89.46 | 89.56 |
| OrthoHash | Hoe et al. (2021) | 75.40 | 80.23 | 82.33 | 69.56 | 75.32 | 77.41 | 73.09 | 75.95 | 76.08 | 87.98 | 90.42 | 90.68 |
| ExchNet$^{\dagger}$ | Cui et al. (2020) | 51.04 | 65.02 | 70.03 | - | - | - | 63.83 | 76.13 | 78.69 | 40.28 | 69.41 | 78.69 |
| A$^2$-Net | Wei et al. (2021a) | 69.03 | 79.15 | 80.29 | 59.60 | 73.59 | 77.69 | 71.48 | 79.11 | 80.06 | 81.04 | 89.34 | 90.75 |
| SEMICON | Shen et al. (2022) | 73.61 | 81.85 | 81.84 | 57.68 | 71.75 | 76.07 | 60.38 | 73.22 | 76.56 | 73.94 | 85.63 | 89.08 |
| **ConceptHash** | (Ours) | **83.45** | **85.27** | **85.50** | **76.41** | **81.28** | **82.16** | **82.76** | **83.54** | **84.05** | **91.70** | **92.60** | **93.01** |

# 4 EXPERIMENTS

**Datasets** We evaluate our method on four fine-grained image retrieval datasets: CUB-200-2011, NABirds, Aircraft, and CARS196. **CUB-200-2011** (Wah et al., 2011) has 200 bird species and 5.9K training/5.7K testing images. **NABirds** (Van Horn et al., 2015) has 555 bird species and 23K training/24K testing images. **FGVC-Aircraft** (Maji et al., 2013) has 100 aircraft variants and 6.6K training/3.3K testing images. **Stanford Cars** (Krause et al., 2013) has 196 car variants and 8.1K training/8.0K testing images. The experiment setting is exactly the same as those in previous works (Cui et al., 2020; Wei et al., 2021a; Shen et al., 2022).

**Implementation details** We use the pre-trained CLIP (Radford et al., 2021) for our image encoder (a ViT/B-32 (Dosovitskiy et al., 2021)) and text encoder (a 12-stacked layers Transformer). The SGD optimizer is adopted with a momentum of 0.9 and a weight decay of 0.0001. The training epoch is 100 and the batch size is 32. We use a cosine decay learning rate scheduler with an initial learning rate of 0.001 and 10 epochs of linear warm-up. We adopt standard data augmentation strategies in training (*i.e.*, random resized crop and random horizontal flip only). For training efficiency, we insert learnable Adapters (Houlsby et al., 2019) to the frozen image encoder (see supplementary material for details). We use the same backbone, adapters, and training setting to fairly compare all the methods.

**Performance metrics** We adopt *mean average precision* which is the mean of average precision scores of the top $R$ retrieved items, denoted as mAP@R. We set $R = $ *full retrieval size* following previous works (Cui et al., 2020; Wei et al., 2021a; Shen et al., 2022).

## 4.1 COMPARISON WITH THE STATE-OF-THE-ART METHODS

For comparative evaluation of ConceptHash, we consider both state-of-the-art coarse-grained (Hash-Net (Cao et al., 2017), DTSH (Wang et al., 2016b), GreedyHash (Su et al., 2018), CSQ (Yuan et al., 2020), DPN (Fan et al., 2020), and OrthoHash (Hoe et al., 2021)) and fine-grained (A$^2$-Net (Wei et al., 2021a) and SEMICON (Shen et al., 2022)) methods. For fair comparisons, the same CLIP pre-trained image encoder (ViT/B-32) is used in all methods for feature extraction. Our implementation is based on the original source code.

**Results.** The fine-grained image retrieval results are reported in Table 1. We make several observations. **(1)** Among all the competitors, our ConceptHash achieves the best accuracy consistently across all the hash code lengths and datasets, particularly in the case of low bits (*e.g.*, 16 bits). This suggests the performance advantage of our method in addition to the code interpretability merit. In particular, it exceeds over the best alternative by a margin of up to 6.82%, 6.85%, 9.67%, and 3.72% on CUB-200-2011, NABirds, Aircraft, and CARS196, respectively. **(2)** Interestingly, previous coarse-grained hashing methods (*e.g.*, OrthoHash) even outperform the latest fine-grained hashing counterparts (*e.g.*, SEMICON). This suggests that their extracted local features are either not discriminative or uncomplimentary to the global features.

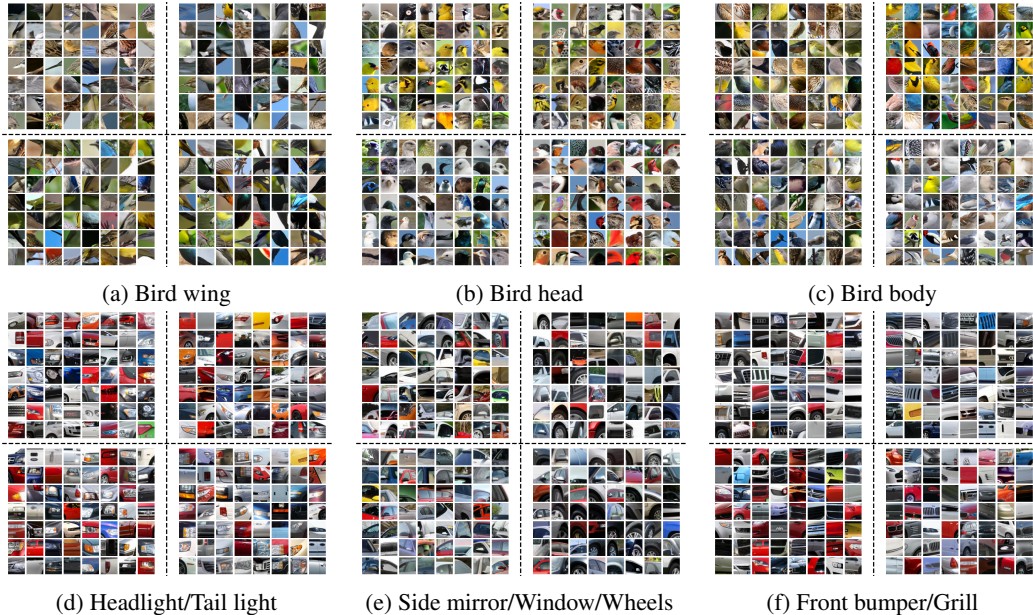

(a) Bird wing          (b) Bird head          (c) Bird body

(d) Headlight/Tail light     (e) Side mirror/Window/Wheels     (f) Front bumper/Grill

Figure 3: We visualize the discovered concepts by our ConceptHash: (a, b, c) The bird body parts discovered on CUB-200-2011. (d, e, f) The car parts discovered on Stanford Cars. Setting: 6-bit hash codes where $M = 3$ concepts are used each for 2-bit sub-code. Bottom-left, top-left, top-right, and bottom-right regions represent the sub-codes 00, 01, 11, and 10 respectively.

## 4.2 INTERPRETABILITY ANALYSIS

To examine *what concepts our ConceptHash can discover*, we start with a simple setting with 3 concepts each for 2-bit sub-code (*i.e.*, totally 6-bit hash code). We train the model on CUB-200-2011 and Standford Cars, respectively. For each concept, we find its attention in the attention map $A$ and crop the corresponding heat regions for visual inspection. As shown in Fig. 3, our model can automatically discover the body parts of a bird (*e.g.*, head, body, and wings) and car parts (*e.g.*, headlight, window, wheels, grill) from the images without detailed part-level annotation. This validates the code interpretability of our method.

**Attention quality.** Although fine-grained hashing methods (*e.g.*, A$^2$-Net (Wei et al., 2021a) and SEMICON (Shen et al., 2022)) lack the code interpretability, local attention has been also adopted. We further evaluate the quality of attention with our ConceptHash and these methods. As shown in Fig. 4, we observe that while A$^2$-Net and SEMICON both can identify some discriminative parts of the target object along with background clutters, our model tends to give more accurate and more clean focus. This is consistent with the numerical comparison in Table 1, qualitatively verifying our method in the ability to identify the class discriminative parts of visually similar object classes.

## 4.3 FURTHER ANALYSIS

**Impact of language guidance.** We evaluate the effect of language guidance (Eq. 4) by comparing with two alternative designs without using the language information: (i) `Random vectors`: Using random orthogonal centers (Hoe et al., 2021) without using visual information; (ii) `Learnable`

Table 2: Effect of language guidance in forming the hash class centers.

| Dataset | CUB-200-2011 | | | NABirds | | | FGVC-Aircraft | | | Stanford Cars | | |
|---|---|---|---|---|---|---|---|---|---|---|---|---|
| Hash centers | 16 | 32 | 64 | 16 | 32 | 64 | 16 | 32 | 64 | 16 | 32 | 64 |
| Random vectors | 77.00 | 79.61 | 82.12 | 71.93 | 73.66 | 76.20 | 71.22 | 77.28 | 79.19 | 88.57 | 87.17 | 88.46 |
| Learnable vectors | 81.55 | 82.39 | 83.86 | 75.80 | 79.76 | 81.66 | 82.08 | 83.43 | 83.62 | 91.03 | 91.92 | 92.84 |
| **Language (Ours)** | **83.45** | **85.27** | **85.50** | **76.41** | **81.28** | **82.16** | **82.76** | **83.54** | **84.05** | **91.70** | **92.60** | **93.01** |

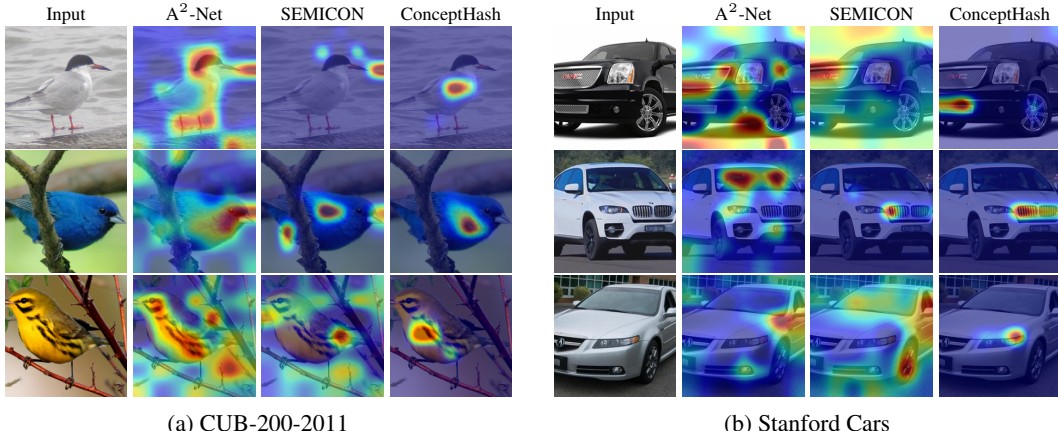

|  (a) CUB-200-2011 | (b) Stanford Cars |

Figure 4: The regions where the hash function will focus on while computing a hash code.

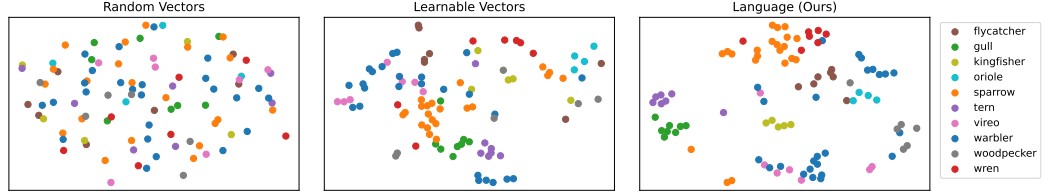

Figure 5: tSNE of the hash centers. The top 10 families of fine-grained classes in CUB-200-2011 are plotted for clarity.

`vectors`: Learning the class centers with discrete labels thus using the visual information. We observe from Table 2 that: **(1)** Our language-guided hash centers yield the best performance, validating our consideration that using extra textual information is useful for visually challenging fine-grained image recognition. **(2)** Among the two compared designs, `Learnable vectors` is significantly superior, suggesting that hash class centers are a critical component in learning to hash and also that imposing the language information into class centers is a good design choice. **(3)** It is worth noting that, even without any language guidance (the `Learnable vectors` row of Table 2), our results are clearly superior to the compared alternatives (see Table 1).

For visual understanding, we plot the distribution of class centers on CUB-200-2011 using the tSNE (Van der Maaten & Hinton, 2008). For clarity, we select the top-10 families of bird species. As shown in Fig. 5, the class centers do present different structures. Using the visual information by `Learnable vectors`, it is seen that some classes under the same family are still farther apart (*e.g.*, the *kingfisher* family, gold colored). This limitation can be mitigated by our language guidance-based design. Furthermore, the class centers present more consistency with the family structures. For example, *tern* and *gull* are both seabirds, staying away from the other non-seabird families. This further validates that the semantic structural information captured by our ConceptHash could be beneficial for object recognition.

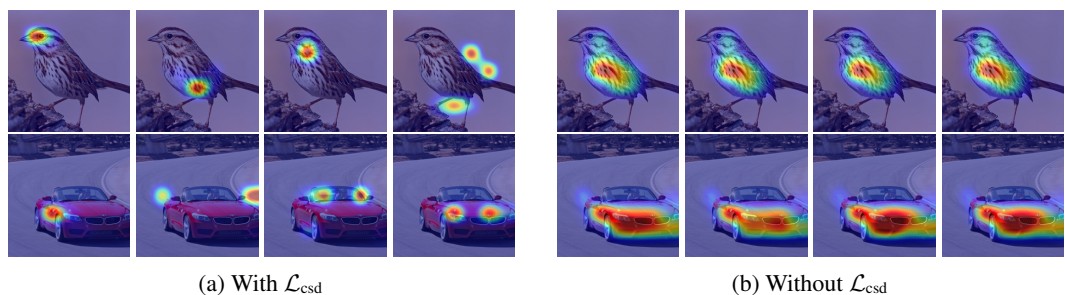

|  (a) With $\mathcal{L}_{csd}$ | (b) Without $\mathcal{L}_{csd}$ |

Figure 6: Effect of concept spatial diversity $\mathcal{L}_{csd}$: The attention maps at the last layer of the Vision Transformer. Setting: $M = 4$.

Table 3: Loss ablation: The effect of adding gradually different loss terms of ConceptHash.

| $\mathcal{L}_{\text{quan}}$ | $\mathcal{L}_{\text{csd}}$ | $\mathcal{L}_{\text{cs}}$ | CUB-200-2011 | | Stanford Cars | |
|---|---|---|---|---|---|---|
| | | | 16 | 64 | 16 | 64 |
| ✗ | ✗ | ✗ | 68.65 | 82.00 | 81.85 | 91.20 |
| ✓ | ✗ | ✗ | 81.12 | 84.14 | 90.03 | 92.72 |
| ✓ | ✓ | ✗ | 81.63 | 84.79 | 90.63 | 92.82 |
| ✓ | ✗ | ✓ | 83.02 | 85.10 | 91.57 | 92.75 |
| ✓ | ✓ | ✓ | **83.45** | **85.50** | **91.70** | **93.01** |

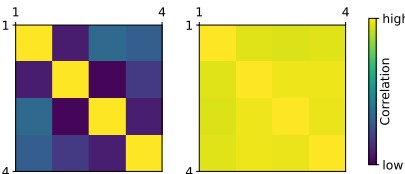

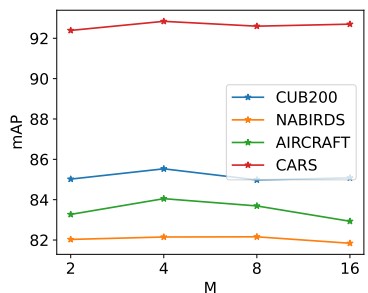

Figure 7: The correlation matrix between attention maps at the last layer of the vision transformer when training with (left) and without (right) the proposed concept spatial diversity constraint $\mathcal{L}_{\text{csd}}$. This is averaged over all training images of CUB-200-2011 with $M = 4$.

Figure 8: Impact of the concept number $M$. Setting: 64 bits.

**Loss design.** We examine the effect of key loss terms. We begin with the baseline loss $\mathcal{L}_{\text{clf}}$ (Eq. 6). We observe in Table 3 that: (1) Without the quantization loss $\mathcal{L}_{\text{quan}}$ (Eq. 7), a significant performance drop occurs, consistent with the conventional findings of learning to hash. (2) Adding the concept spatial diversity constraint $\mathcal{L}_{\text{csd}}$ (Eq. 8) is helpful, confirming our consideration on the scattering property of underlying concepts. We find that this term helps to reduce the redundancy of attention (see Fig. 6 and Fig. 7). (3) Using the concept discrimination loss $\mathcal{L}_{\text{cd}}$ (Eq. 9) further improves the performance, as it can increase the discriminativeness of the extracted concepts.

## 5 CONCLUSION

In this work, we have introduced a novel concept-based fine-grained hashing method called ConceptHash. This method is characterized by learning to hash with sub-code level interpretability, along with leveraging language as extra knowledge source for compensating the limited visual information. Without manual part labels, it is shown that our method can identify meaningful object parts, such as head/body/wing for birds and headlight/wheel/bumper for cars. Extensive experiments show that our ConceptHash achieves superior retrieval performance compared to existing art methods, in addition to the unique code interpretability.

**Limitations** It is noted that increase in the number of concepts $M$ can lead to overfitting and negatively impact interpretability, resulting in attention maps being scattered randomly around (see Fig. 8 and Fig. 9). The discovered concepts require manual inspection as the general clustering methods. Addressing these limitations will be the focus of our future work.

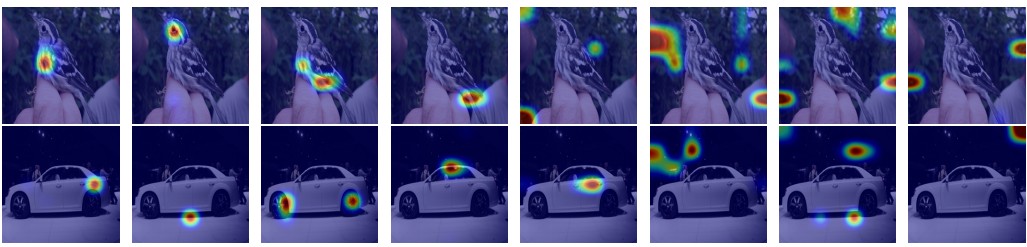

Figure 9: Example attention maps. Setting: $M = 8$.

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

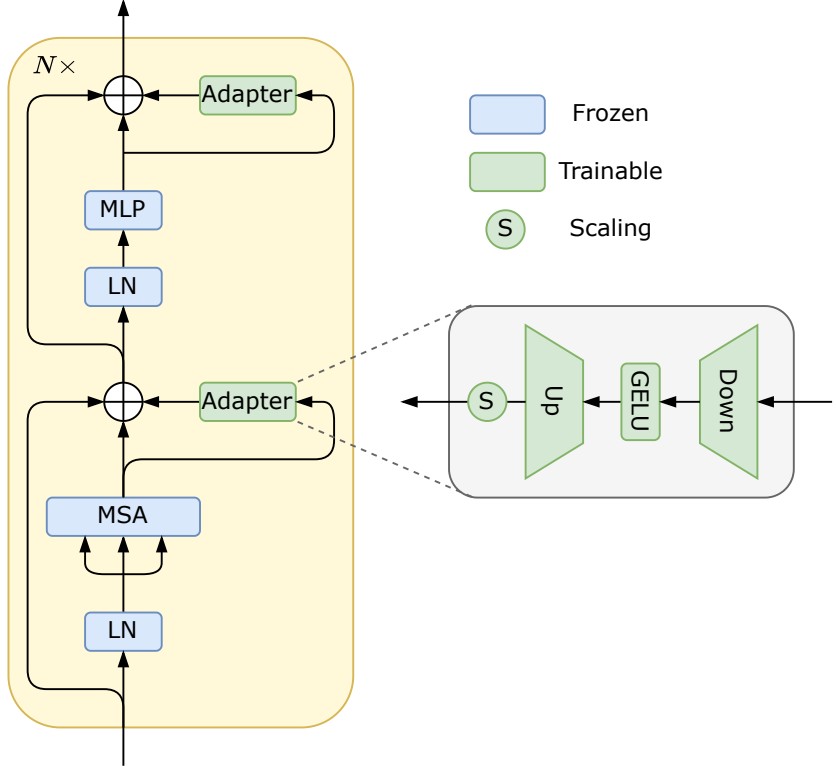

Figure 10: Two adapters are added after the multi-head self-attention layer (MSA) and the feedforward network (MLP). LN denotes layer normalization for each block of a standard vision transformer.

## A    IMPLEMENTATION OF ADAPTER

To increase training efficiency, we add adapters to the vision transformer instead of fine-tuning all parameters. We adopt the architecture in AdaptFormer Chen et al. (2022a) and define our adapter as:

$$\text{adapter}(z) = s \cdot W_{\text{up}} \cdot \text{GELU}(W_{\text{down}} \cdot \text{LN}(z)), \tag{10}$$

where LN is a layer normalization layer Ba et al. (2016), $W_{\text{down}} \in \mathbb{R}^{D_{\text{down}} \times D}$ is the weights of down projection and $W_{\text{up}} \in \mathbb{R}^{D \times D_{\text{down}}}$ is the weights of up projection, GELU is the non-linear activation function Hendrycks & Gimpel (2016), and $s \in \mathbb{R}$ is a learnable scaling factor. $D_{\text{down}}$ is set as $384$.

We added two adapters for each block of the vision transformer, one after multi-head self-attention (MSA) layer and one after feedforward network (MLP). The output of $l$-th block of the vision transformer is computed as:

$$\hat{Z}^{(l)} = \text{MSA}(\text{LN}(Z^{(l-1)})),$$

$$\hat{\hat{Z}}^{(l)} = \text{adapter}(\hat{Z}^{(l)}) + \hat{Z}^{(l)} + Z^{(l-1)},$$

$$\widetilde{Z}^{(l)} = \text{MLP}(\text{LN}(\hat{\hat{Z}}^{(l)})),$$

$$Z^{(l)} = \text{adapter}(\widetilde{Z}^{(l)}) + \widetilde{Z}^{(l)} + \hat{\hat{Z}}^{(l)}. \tag{11}$$

See Fig. 10 for the detail of the computational graph. We follow Houlsby et al. (2019) to insert our adapters.

## B    RETRIEVAL ON FAMILY SPECIES

In this section, we evaluate the methods by replacing the fine-grained labels with family labels in order to assess the semantic ability of the hash codes. The CUB-200-2011 dataset is chosen

Table 4: Performance (mean average precision) of retrieval by family species on CUB-200-2011.

| Methods | CUB-200-2011 | | |
|---|---|---|---|
| | 16 | 32 | 64 |
| ITQ Gong et al. (2012) | 20.00 | 23.46 | 27.09 |
| HashNet Cao et al. (2017) | 24.40 | 35.62 | 38.13 |
| DTSH Wang et al. (2016b) | 36.96 | 37.81 | 39.49 |
| GreedyHash Su et al. (2018) | 44.46 | 55.62 | 60.98 |
| CSQ Yuan et al. (2020) | 31.62 | 34.47 | 35.25 |
| DPN Fan et al. (2020) | 34.09 | 36.28 | 36.84 |
| OrthoHash Hoe et al. (2021) | 34.16 | 36.95 | 37.61 |
| $A^2$-Net Wei et al. (2021a) | 45.62 | 50.93 | 52.95 |
| SEMICON Shen et al. (2022) | 43.10 | 53.24 | 56.80 |
| ConceptHash (Ours) | **60.54** | **63.44** | **67.20** |

as the benchmark. Table 4 presents two key observations: (i) Our ConceptHash outperforms previous methods by a significant margin, highlighting the effectiveness of our approach. This result underscores the superiority of our methods in capturing the semantic information encoded within the hash codes. (ii) Random-center-based hashing methods like CSQ Yuan et al. (2020) perform worse than older hashing methods such as DTSH Wang et al. (2016b), even though they outperform them in fine-grained retrieval (Table 1 in the main paper). A likely explanation is that the training objective of random-center-based hashing primarily focuses on learning to generate the fixed target hash codes, thereby ignoring the semantic relationships (such as family information) between the fine-grained classes.

