# OpenReview forum: "ConceptHash: Interpretable Fine-Grained Hashing with Concept Discovery"
_ICLR.cc/2024/Conference — ICLR 2024 Conference Withdrawn Submission_

### Official Review · Reviewer_7pJE · 2023-10-31

**Soundness:** 3 good
**Presentation:** 3 good
**Contribution:** 2 fair
**Rating:** 5
**Confidence:** 4

**Summary:**

This paper introduces an image hashing method called ConceptHash. ConceptHash is designed to automatically discover human-understandable concepts within images. It leverages a pretrained vision-language model to achieve semantic understanding for sub-codes, resulting in strong performance on multiple fine-grained image retrieval benchmarks.

**Strengths:**

1. The motivation behind this work is clearly elucidated, emphasizing the importance of enhancing the interpretability of hash codes.

2. The method's description is well-structured and the results surpass those of competing methods.

**Weaknesses:**

1. The title of the paper might be somewhat confusing (the title in OpenReview and the paper is different). After reading the paper, it becomes apparent that the proposed method generates hash codes for fine-grained image retrieval, which aligns more with "retrieval" than "generation." Therefore, the title could be revised to better reflect its purpose and avoid misleading readers into thinking it's a generation task.

2. While interpretability of hash codes is crucial for fine-grained retrieval, the proposed method may not be considered highly novel. For instance, a previous work [1] also introduced concept-level interpretability for hash codes, although they referred to it as "instance-level" in their paper. The primary distinction between the proposed method and [1] lies in ConceptHash's utilization of CLIP to obtain semantic information, as opposed to explicitly generating object proposals.

[1] Instance-aware hashing for multi-label image retrieval. TIP, 2016.

3. The utilization of pretrained vision-language models, such as CLIP, for enhancing image retrieval is a common approach currently. The authors assert that “…CLIP … ensures that our learned hash codes are not only discriminative within fine-grained object classes but also semantically coherent.” However, CLIP is pretrained on image-text pairs, and the model processes image/text as a whole during learning, potentially emphasizing coarse-grained knowledge. Therefore, it would be beneficial for the paper to clarify how the proposed method addresses the challenges of fine-grained hashing and specify its uniqueness compared to the existing utilization of CLIP in image retrieval.

**Questions:**

1. Figure 3 could benefit from larger-sized images, and it might be more effective to show only a selection of 3-5 examples, while additional examples can be included in the supplementary material.

2. The heat map visualization in Figure 4 may be somewhat perplexing. While ConceptHash can focus on a small part of an object, such as the headlight corner of a car in the last image, it raises questions about whether focusing solely on this small area is sufficient for distinguishing image effectively.

---

### Official Review · Reviewer_nLVP · 2023-10-31

**Soundness:** 2 fair
**Presentation:** 3 good
**Contribution:** 2 fair
**Rating:** 5
**Confidence:** 4

**Summary:**

This paper introduces ConceptHash, an approach designed to achieve interpretability of fine-grained hashes. The framework is predicated on the Vision Transformer (ViT) and ingests learnable concept tokens alongside image patches as inputs. Each query token in the output corresponds to a subcode, which are concatenated to form the final hash code. Additionally, language guidance of a pretrained vision-language model is also incorporated, ensuring the semantic consistency of the learned hash codes.

**Strengths:**

The paper is well writing and its goal is clear to the reader.

The experimental results validate the effectiveness of the proposed method, and the performance of the proposed method is significantly better than the state-of-the-art method.

**Weaknesses:**

The automatic discovery of human-understandable concepts (sub-codes) is a critical aspect of ConceptHash. However, it is an implicit learning process, the robustness of this process remains an open question. How sensitive is it to variations in training data or domain-specific nuances?

The motivation of introducing learnable concept tokens with image patch tokens seems rational. However, it lacks an explicit experimental validation to substantiate the initial premise.

Certain aspects of the paper lack rigor, such as the role for a learnable $ \hat{W}_{c}$ in the fourth loss function and the interpretation of the ablation study pertaining to it.

Fairness of comparative experiments：1）Contrasting with methodologies like SEMICON and A2-Net that employ a CNN as the backbone, this paper leverages a pre-trained CLIP model. A noteworthy divergence in performance is observed between the two approaches. 2）Given the extensive training data of CLIP, which likely encompasses fine-grained data such as CUB200, the effectiveness of the idea proposed in this paper warrants further validation, such as through the substitution of different backbone architectures.

**Questions:**

Refer to the weakness

---

### Official Review · Reviewer_cLQM · 2023-11-01

**Soundness:** 3 good
**Presentation:** 3 good
**Contribution:** 3 good
**Rating:** 6
**Confidence:** 3

**Summary:**

The article presents ConceptHash, an algorithm that maps sub-codes to human-understandable concepts for interpretable hashing. ConceptHash achieves this by:
1. Learning concept tokens appended to the original mage one. These concept tokens are processed via a learnable hash function after being shifted by learned specificity embeddings.
2. Language-guidance vectors extracted from the class names of the original dataset and projected through the hash code space.
3. Employing four objective functions which enforce a) correct classification of the hash code; b) correct classification of the concept representation; c) low quantization error from binarized language embeddings and hash codes; d) spatial diversity of the concept tokens (i.e. making their attention focus on a different part of the image).

Experiments on various fine-grained datasets show that the method outperforms by a margin all competitors while providing concept tokens that are interpretable after cropping their most attended image parts.

**Strengths:**

1. The approach is a relatively simple idea (i.e. using concept tokens to derive hash codes, making them interpretable via language/class guidance) that works very well in practice, with gaps of more than 10% in mean average precision for 16-bit codes (Table 1). These results show that ConceptHash is a competitive method that future works may use a strong baseline for fine-grained dataset hashing, especially when using pretrained vision-language models as backbones.

2. The article is well-written and easy to follow. The related work (Section 2) covers extensively the literature on the topic while clarifying the contribution of the paper. The terminology and notation of Section 3 are clear and not ambiguous, making it easy for the reader to understand the technical components. Section 4 always describes the motivation behind each experiment, and provides extensive quantitative and qualitative analyses on the method itself.

3. The method focuses on the interpretability of the sub-codes. The visual maps of Fig. 4 and Fig. 6 qualitatively show that the obtained concept maps point to a more semantically meaningful part of the image (w.r.t. previous work) and cropping and visualizing the learned concept tokens can provide insights on the actual semantic part that each token focuses on (Fig. 3).

4. I appreciated the analyses of Table 2, showing how semantic guidance (i.e. a standard classifier) is the key to the performance while language guidance itself (e.g. coming from CLIP) boosts them a bit. This shows that, potentially, a non-VLM model may be used to implement ConceptHash.

**Weaknesses:**

1. While the method provides interpretable tokens, the human interpretability is given by the visualization of relevant image crops (Fig. 3). This is not a problem per-se but becomes such when the number of concept tokens may lead to overfitting, thus making the tokens not interpretable. The manuscript fairly points to this weakness in Section 5 and Fig. 9.
2. Following on the previous, analyses on interpretability are held only qualitatively. To strengthen the claim of better interpretability, it would have been helpful to conduct quantitative studies, either as user studies (e.g. whether two images are of the same category based on the highlighted regions) or using explainability metrics (e.g. remove and classify) and/or using part annotations available (e.g. in CUB) to verify if parts of the birds are localizable via the learned concepts. At the moment, the reader has to assess the interpretability via the few available examples and, while they are indeed interesting and explicit, quantitative analyses would confirm the consistency in achieving better interpretability.
3. From the article, I did not find clear how the hash function $h$ is implemented. For clarity, it would be helpful to expand on it (e.g. after Eq. (3)).
4. There is no ablation on the use of adapters (Appendix A). While they are not the main contribution of the paper, it would have been helpful to show if the performance is affected by the particular choice of adapters for both the presented method and the baselines (as fine-tuning the backbones may lead to overfitting and worsen their results). Similarly, the impact of the specificity embeddings is unclear as not quantitatively ablated at the moment.
5. In principle $L_{cls}$, $L_{quan}$, and $L_{cd}$ are "redundant" as they all provide semantic supervision, even if at different levels. Table 3 at the moments shows how $L_{quan}$ and $L_{cd}$ interact, but it does not show the impact of $L_{cls}$ and its interaction with the other losses (e.g. considering two of them at the time). It would be helpful to expand the table to provide the full picture.
6. From the results/benchmarks, it is not clear if ConceptHash generalizes to classes not present during training. While it would be interesting to provide quantitative analyses on this, even a discussion on potential generalization issues would help in expanding the possible directions for future work and provide insights on the model.

Minor:
- $L_{cs}$ vs $L_{cd}$ in Tab. 3.
- Red font at the end of Section 1.

**Questions:**

1. Is it possible to quantify the interpretability of the obtained concept tokens w.r.t. existing approaches?
2. How is $h$ implemented?
3. What is the impact of the adapters for the proposed method? Could it impact also the baselines?
4. What is the impact of the specificity embeddings?
5. How do the semantic losses interact (i.e. how would the full ablation in Tab. 5 look like)?
6. Does the model generalize to unseen classes?

**Details Of Ethics Concerns:**

None.

---

### Official Review · Reviewer_cpFr · 2023-11-09

**Soundness:** 3 good
**Presentation:** 3 good
**Contribution:** 3 good
**Rating:** 5
**Confidence:** 3

**Summary:**

The authors propose a fine-grained hashing approach ConceptHash, desighed for interpretability. Previous work derive hash codes from mixed local and global features, which lacks the code interpretability. In ConceptHash, the hash codes are derived from designed learnable concept tokens, offering sub-code level interpretability. Extensive experiments on fine-grained image retrieval benchmarks demonstrate the effectiveness of ConceptHash.

**Strengths:**

1.	ConceptHash offers unique sub-code interpretability, where each sub-code is associated with a specific visual concept.
2.	ConceptHash achieves impressive results on four fine-grained image retrieval benchmarks.
3.	The ablations are clear and well demonstrate the effectiveness of proposed learning objectives and the language guidance.

**Weaknesses:**

1.	In Fig.3, it shows that different concept tokens clearly focus on different visual concepts. However, the meaning of each sub-code is unclear for me. For example, I can’t distinguish the difference of (00,01,10,11) in Fig.3 (a). Would more bits for each concept help to distinguish the sub-code?
2.	In Fig.4, I found that ConceptHash well attention to the object, while the attention region seems not discriminative.

Others:
1.	The title is “Interpretable hashing for fine-grained retrieval and generation”, but there is no “generation” content in this paper.
2.	In Eq.3, should the dimension of E_m be (1, D) ?
3.	In Eq.7, b_n  b_i ?
4.	Typo: Sec. 4.2, an extra 'A' in line 3.

**Questions:**

Please find my comments on the weaknesses.